# Chagas Disease: Medical and ECG Related Findings in an Indigenous Population in Colombia

**DOI:** 10.3390/tropicalmed8060297

**Published:** 2023-05-29

**Authors:** Simone Kann, Gustavo Andrés Concha Mendoza, Maria Hartmann, Hagen Frickmann, Lothar Kreienbrock

**Affiliations:** 1Medmissio, 97074 Wuerzburg, Germany; 2Organisation Wiwa Yugumaiun Bunkuanarrua Tayrona (OWYBT), Valledupar 200001, Colombia; 3Department of Biometry, Epidemiology and Information Processing, WHO Collaborating Centre for Research and Training for Health in the Human-Animal-Environment Interface, University for Veterinary Medicine Hannover, 30559 Hannover, Germany; 4Institute for Medical Microbiology, Virology and Hygiene, University Medicine Rostock, 18057 Rostock, Germany; 5Department of Microbiology and Hospital Hygiene, Bundeswehr Hospital, 20359 Hamburg, Germany

**Keywords:** Chagas disease, cardiomyopathy, rhythm disturbances, sudden heart death

## Abstract

Chagas Disease (CD) is highly prevalent among the indigenous populations in the Sierra Nevada de Santa Marta, Colombia. Villages examined show prevalence rates ranging from 43.6% up to 67.4%. In the present study, associated medical conditions were assessed with a particular focus on ECG alterations. CD diagnosis was based on a rapid test, two different ELISAs, and a specific and highly sensitive Chagas real-time PCR. In both CD positive and CD negative patients, relations of the status and medical (physical examination-based, questionnaire-based) and/or electrocardiogram-based findings were investigated. As expected, CD-associated symptoms and complaints were predominantly found in CD-positive patients. Interestingly, ECG-findings were found to show the potential of leading to early CD diagnosis because ECG alterations were already seen in early stagechanges of the disease. In conclusion, although the observed ECG changes are unspecific, they should be considered as an indicator for a CD screening and, in case of positive results, an associated early treatment of the disease.

## 1. Introduction

Chagas disease (CD) is caused by the protozoan parasite *Trypanosoma cruzi* (Chagas, 1909), (Kinetoplastida, Trypanosomatidae). CD-transmitting triatomines (Hemiptera, Reduviidae, Triatominae) are found in Central and South America, Mexico, and the southern United States [1,2,3,4]. The examined communities of the indigenous population called Wiwas live in the Sierra Nevada de Santa Marta, Colombia and show a very high burden of the disease. In some of their villages, even higher prevalence rates were found than, for example, in Bolivia [5,6], which is considered to be one of the countries with the globally highest CD burden.

The local main mode of transmission in the communities is via Triatomines that suck human blood, thereby leaving infected stool close to the stich point. Thereafter, an itching sensation followed by scratching leads to the inoculation of the pathogens into the body. Incorporation of the infectious agents via mucous membranes as well as vertical transmission are also common, next to infections due to contaminated food. In addition, infections driven by blood or organ donations are feasible but rare in Wiwa communities.

The region examined has many potentially transmitting vectors: One of the main vectors in the area examined is *Rhodnius prolixus*. It is a member of the family Reduviidae and transmits CD to humans and other mammals. *R. prolixus* is usually present in rural areas and feeds on the blood of animals such as birds, rodents, and opossums. It is attracted by the carbon dioxide that mammals exhale and is typically active at night when the hosts are asleep [7,8]. As the Wiwas live close together with their livestock and as the housing and climate conditions are favorable for the Triatomines (palm roofs, mud walls, etc.), transmission to humans is common [9].

Another locally highly prevalent triatomine is *Triatoma dimidiata*, which transmits CD as well. It is also attracted to carbon dioxide, but in addition to heat and moisture, showing its perfect adaption to the climate conditions found in the Sierra Nevada de Santa Marta. Also, other triatomes can be observed in the region, for example, *T. maculata*, *T. infestans*, *R. pallenscens* (the so-called “palm tree triatomine”), and *Panstrongylus geniculatus*; however, those play only a subordinate role so far [8,10,11].

Other reasons for the observed high number of infections in the indigenous populations, next to the poor socio-economic living conditions, comprise a lack of countermeasures like fumigations, insufficient knowledge and awareness of the disease, and sparse access to surveillance and prevention programs, for example, due to their retracted living places. Health care is also limited and just consists of a nurse in a health point with irregular consultation days without options for diagnosing *T. cruzi*-infections.

Once a patient is infected, two stages of the disease usually occur. Patients in the acute stage typically present with flu-like symptoms. Because these symptoms are unspecific, the distances to reach the next physician are long for the Wiwas, and the complaints usually disappear without therapy after a certain time, the acute CD stage often remains undetected. Even if Wiwa patients would consult a physician early, awareness would be missing as the availability of diagnostic tests was not in place. This is a matter of concern because the acute stage is the most promising regarding favorable therapeutic outcomes [12,13]. In rare cases, especially seen in young children, the acute stage can be associated with severe complications like myocarditis, encephalitis, and others, which go along with a high fatality rate [14,15].

After the acute stage, the patient enters into a chronic stage. About 30–40% of the chronically infected patients develop complications. These can be related to the heart (cardiomyopathy, rhythm disorders, sudden heart death, etc.) and/or to the gastrointestinal tract (megacolon, megaesophagus, etc.). It can take decades until the final stage is reached; however, in all cases with complications, lifetime and quality are impaired, resulting in high mortality and morbidity rates [16,17] as well as premature death [14].

In the highly endemic area, where the Wiwas live, very high proportions of infected individuals are already in the chronic stage of CD at the time of diagnosis as described previously [18]. The main manifestations seen in the indigenous populations are heart-related. It is known that in CD, ECG findings are mostly unspecific [15]; however, combined with the knowledge of a person living in a high endemic area, they might be helpful in supporting the initiation of early diagnosis.

Accordingly, the purpose of this study was to investigate (1) the regional prevalence of CD, (2) if signs and symptoms gained by physical examinations and/or questionnaires can be used as an indicator for a CD diagnostic approach, (3) what ECG findings were present in the indigenous communities in CD positive and CD negative participants, (4) if a (specific) ECG finding should lead to further diagnostics in high endemic regions for CD, and (5) if the ECG should be included in CD screenings in the indigenous communities/persons living in a high-risk area in general.

## 2. Materials and Methods

### 2.1. Ethical Clearance

Both parts of the study were performed in accordance with the principles of the Declaration of Helsinki. The study providing the data for part I (2017–2019) was approved by the Ethics Committee of Santa Marta, Colombia (Acta No. 032018). The study providing the data for part II was approved by the Institutional Ethic Committee for Investigation of Bogota, Colombia (Acta No. 2019-4). Written informed consent was obtained from each participant or from a parent or legal guardian of a child prior to participation.

### 2.2. Study Design and Population

The data and samples were collected during the course of two studies. The first study, “Program against Chagas Disease in the Indigenous Population of Colombia”, was conducted from July 2017 to March 2019, included 684 volunteers, and is referred to as “part I” in the following. The second study, “Colombia-Germany research program on diagnostics, research, treatment and prevention of Chagas Disease and Emerging Infectious Diseases in vulnerable groups”, started in February 2020 and included 450 volunteers. It is referred to as “part II”.

In both parts of the assessment, the procedure of collecting information and samples were basically the same. Indigenous people from the Wiwa tribe were asked in nine villages for their voluntary participation. The number of inhabitants screened (sample population n/participation rate in %) was as followed for the different study sites: Tezhumake 173/69.2%, Seminke 110/70.9%, Cherua 94/78.3%, Ashintukwa 106/42.4%, Ahuyamal 86/87.5%, Surimena 63/92.6%, Sabannah de Higuieron 86/63.2%, Dungkare 38/49.8%, and Potrerito 45/40.5%. Of note, Ashintukwa and Potrerito are transit cities and so, there is only little permanent population. In total, data from 1134 Wiwas was collected.

All villages were chosen because of an assumed high prevalence for CD because they are close to previously tested areas with high CD prevalence [18] in the north-east of Colombia, the Sierra Nevada de Santa Marta.

### 2.3. General Screening Design

In all nine villages, the following procedure was performed: Adults defined as being 12 years of age or older were initially tested with a Chagas rapid test targeting anti-*T. cruzi* antibodies. If the RT was positive, serum was taken. The serum was used to additionally perform two different ELISAs and a Chagas-RT-PCR. In children below 12 years of age, serum was taken directly and the RT, the two ELISAs and the PCRs were performed in all instances.

As a control group, individuals were included whose serum was consistently negative in the RT, ELISAs, and PCRs. These were 42 person groups in Ahuyamal and 34 person groups in Surimena. For this population as well, the full diagnostic procedure as described below was performed, independent of their age.

Most of the volunteers in all groups answered a questionnaire and received a full physical examination and an ECG. The questionnaire asked for general personal data, medical history, actual complaints, allergies, and diseases which constitute contraindications for the medical treatment of CD. The section on CD disease-related queries asked specifically for cardiac symptoms (pressure on the chest, chest pain), gastrointestinal symptoms (regurgitation, heart burn, dysphagia), dizziness, syncope, dyspnea, swollen lymphatic nodes, fatigue, and exhaustion. Illiterates received assistance by an educated promotor, parents filled out the questionnaire for their children.

All volunteers were informed about their individual study results by a physician and were registered—if CD positive—in the official Colombian data base of the health care provider Dusakawi in line with Colombian guidelines. If the CD test result was positive, access to treatment was guaranteed. In particular, a drug observed treatment was offered for CD positively tested patients directly after the screening process within the study. Patients who could not participate directly, for example, due to pregnancy, were entitled for treatment at any other later timepoint, coordinated by Dusakawi.

### 2.4. Laboratory Diagnostics

Serology was performed using a Chagas Rapid Test (Chagas AB Rapid, Standard Diagnostics Inc. Bioline, Bogota, Colombia) and two different ELISAs (Chagatest ELISA recombinant v. 4.0 Wiener Lab, ELISA-Recombinante and Chagas IgG ELISA Lisado IBL, Wiener Lab, ELISA-Lisado) according to the manufacturers’ protocols. The choice of the two different ELISAs was performed in accordance with Colombian guidelines for the diagnosis of CD, which are based on WHO recommendations [19].

DNA extraction and real-time PCR were performed as described previously by Kann et al. (2022) [20]. In short, extractions were made from serum following the manufacturers’ protocols of the RTP Pathogen Kit (Invitek Molecular GmbH, Berlin, Germany) with the samples from study part I and of the MagaBio Plus Virus DNA/RNA purification kit version 2 (Hangzhou Bioer Technology Co., Ltd., Hangzhou, China) for the samples from study part II. The extraction and elution volumes were in a similar range over the compared extractions schemes with 200 µL and 60 µL for the RTP assay and 300 µL and 80 µL for the MagaBio assay, respectively [20].

The NDO-(newly developed one)-PCR (patented) was used, following the protocol as described in a previous publication [9], for study part I. For study part II, the now commercially available Chagas kit from TibMolBiol, Berlin, Germany: *T. cruzi* LightMix^®^, Ref. 53-0755-96, PhHV Extraction Ctrl. Ref. 66-0901-96, Lyophilized 1-step RT-PCR Polymerase Mix, Cat-No 90-9999-96), was applied. As a minor modification compared to the previous protocol, the probe sequence had been slightly altered to 5′-TCG+AACCCC+ACCTCC-3′, the “+” symbol marks locked nucleic acid (LNA) bases included to alter the annealing temperature. A synthetically produced positive control and a *T. cruzi* strain Tulahuen-based positive control were used in parallel. All primers and probes were purchased from Eurofins MWG. All runs were performed on a RotorGene Q cycler (Qiagen, Hilden, Germany).

### 2.5. Transport

Samples arrived in a cooling box the same day they had been taken at the laboratory (Laboratorio Salud Publica) in Valledupar, Colombia, and were then either processed directly or stored at −20 °C. Serologic tests, RT, and extractions were made at the same place. The extracts were used for PCR diagnostics and thereafter stored at −20 °C until their transfer on dry ice to Germany. The transport was conducted in line with all required guidelines and airfreight requirements and processed by a specialized company called World Courier (Frankfurt, Germany). After arrival in Germany, sera and extracts were stored at −80 °C until their further use.

### 2.6. CD Classification

Acute, newly infected cases were defined by the direct detection of the parasite nucleic acids in the blood, that is, by a positive PCR result, and being negative in all serological tests.

To detect CD in its chronic phase, two different serological techniques are recommended by the WHO and demanded by the Colombian guidelines. If both required ELISAs were positive, the patient was considered chronically infected. A third ELISA would have been required in case of discordant results; however, as discordant results did not occur in this study, a third ELISA was not needed. In addition, we performed a Chagas antibody-specific RT.

Chronic cases were furthermore divided into patients being positive in the two ELISAS (and the RT) and (a) negative by Chagas PCR or (b) positive by Chagas PCR. The last group was interpreted as acutely re-infected, re-activated, and/or as an (early) chronic stage of CD infection.

### 2.7. Electrocardiograms (ECG)

A 12-chanel ECG (Germar GMG S.A.S, SE-3 DE, Bogota, Colombia) were performed by a trained person, and specific electrodes for adults and children were used. The ECGs were interpreted by the study physician, and a re-evaluation was performed by a specialist for internal medicine.

Chagas positive patients’ ECGs were classified applying the Chagas associated American Heart Association (AHA) criteria [15]. A requirement for using this classification is at least a CD positive serology. However, to better compare ECG results of CD positive patients with results of CD negative patients and to distinguish between CD-related and CD-non-related findings, the same classification was also used for ECG results of CD negative volunteers.

The ECGs were classified following the AHA classification as published in Nunes et al. [15]. In short, the chronic stage was distinguished into the indeterminate stage (“A”) and the Chagas cardiomyopathy stage (“B1-D”). Classification “A” describes patients at risk for the development of heart failure, thereby, serology is positive, but neither structural cardiopathy nor heart failure symptoms are present. The ECG is normal, the individuum does not show digestive alterations. “B1” characterizes patients suffering from structural cardiopathy with evidence in ECG by typical changes, but patients still have a normal global ventricular function and show neither current nor previous signs and symptoms of heart failure. “B2” describes patients with Chagas-associated dilated cardiomyopathy/heart failure, including ones with structural cardiopathy characterized by global ventricular dysfunction but neither current nor previous signs and symptoms of a heart failure. “C” classifies patients with ventricular dysfunction and current or previous symptoms of heart failure (New York Heart Association functional class NYHA FC I, II, III, or IV). “D” comprises affected patients with refractory symptoms of heart failure at rest despite optimized clinical treatment, requiring specialized interventions. Arrhythmia and conduction system-related disease can occur from classification “B1” to “D”.

### 2.8. Data Management and Statistical Analyses

The gained data was transferred into the statistical analysis program SAS (version 9.4 TS level 1M5, SAS Institute Inc., Cary, NC, USA) for further statistical assessment. Data analysis was performed both based on the CD status as defined above as well as in a dichotomous way comparing the outcomes of CD-negative and CD-positive study participants only.

For checking similarity of self-reported complaints versus medical practitioners’ clinical diagnoses, Cohen’s kappa was calculated. Classical χ^2^-tests were used to check for homogeneity of CD-status within the different associations. To investigate the relation between CD-status and quantitative variables under assessment, ANOVA-models were used taking age as continuous parameter into account for adjustment. For categorical variables, a multifactorial logistic regression was used with age categories. As an exploratory error level, 5% was accepted; a multiple adjustment was not applied.

## 3. Results

### 3.1. Demographic Data

In total, we received data from 1134 volunteers. This includes clinical data (questionnaires and physical exams) with or without ECG records. For a better overview on the structure of the sample collective, please see Figure 1.

From 1134 volunteers in nine villages, 777 were 12 years and older, whereas 353 were below 12 years of age. For four volunteers, the information about their age was missing. In line with the age-related protocol, questionnaire data from 801 volunteers was available. Overall, 406 (51%) observations were from female volunteers and 395 (49%) were from male volunteers, 344 (43%) persons were younger than 12 years, and 457 (57%) persons were 12 years and older. The median age of the volunteers below 12 years was 7.0 years, the distribution of participants being 12 and more years-of-age old was slightly right-skewed with a median of 30.8 years, a 95% percentile of 65 and a maximum age of 89.5 years with only minor differences between male and female participants.

### 3.2. Chagas Testing

For 801 volunteers, both Chagas testing and questionnaire information was available (see blue sub-population in Figure 1). Within this group, 305 were positive for CD serologically only, 97 serologically and by PCR, 2 persons showed seronegative results and positive PCR results, and 397 were negative in all tests.

In the age group 12 years and younger, only 5 female and 11 male CD cases occurred, whereas in the elderly group (12 years and older), 209 women and 179 men were positive for CD.

CD-findings varied between 67.4% and 43.6% in the villages, indicating the area to be a high-prevalence region (summary in Table 1). The overall difference between the villages was statistically significant in the general χ^2^-test (*p* = 0.0131).

Within the sub-population of CD-positive individuals, an additional split of serologically positive cases into special regional patters was performed, thereby discriminating PCR positive cases, showing *T. cruzi* to be circulating in the blood stream, from PCR negative cases. As an example, Cherua showed 5 PCR positive cases and 39 cases being just positive in serology. This analysis resulted in an active CD rate as defined by combined serology-positive and PCR-positive cases of 11.4%. In contrast, Potrerito had 11 PCR-positive cases among 28 CD-positive cases, leading to a rate of 39.3%. Thus, the assessment demonstrated both a different general level of prevalence in the villages as well as a different ratio of infections with actively circulating *T. cruzi* in the bloodstream.

### 3.3. Results Obtained from the Questionnaires and the Medical Examinations

No significant relation between sex and CD-status was found, but CD prevalence differed between the age groups (see Figure 1). Measured body weight and size as well as diastolic and systolic blood measurement and their relation to CD-status were not presented here due to their strong confounding effect related to age.

A total of 243 persons described one or more medical problems. Quantitatively dominating symptoms were stomach complaints with *n* = 116 cases (28.9%), dizziness with *n* = 86 cases (20.5%), chest pain with *n* = 61 cases (13.9%), fatigue with *n* = 37 cases (8.3%), swollen lymph nodes with *n* = 25 cases (5.9%), and shortness of breath with *n =* 17 cases (4.1%), while syncope was reported by only one person.

Relating these complaints to CD-status and two major age-groups, the data suggests an increased association of complaints and CD-status with an odds ratio of 3.86 for participants younger than 12 years and of 2.24 for participants being 12 years of age and older (see Table 2).

Medical diagnoses were recorded in the course of the examination conducted by the study’s physician. Thereby, most diagnoses were not related to CD, but to gastro-intestinal and respiratory diseases, which frequently occur in the Wiwa population as published before [21]. In four cases, cardiac complaints were found: 1 case with NYHA IV cardiac dysfunction, 1 acute myocardial infarction, 1 heart murmur, 1 bradycardia. All patients were positive for CD. These findings were further evaluated, performing similarity analyses comparing self-reported complaints with the medical practitioner’s diagnoses. Thereby, similarities were not detected with kappa-measures close to zero (data not shown in detail).

### 3.4. Results from Electrocardiograms (ECG)

Children being 12 years of age and younger did not show pathologic findings in 95.7% of the cases. Eleven children showed irregularities in the ECG, however, all of them were CD-negative. Therefore, this sub-collective was not described in more detail in the further course of the analyses.

The majority of pathological ECG findings were associated with patients being 12 years of age and above, comprising 72.7% in category “A”, 19.1% in “B1”, 5.7% in “B2”, 2.2% in “C”, and 0.2% in “D” independently from the CD status. The proportions of the different categories were significantly associated with CD-status with a strong shift into increased ECG-AHA classification within the CD-positive group of participants being 12 years and older (*p* = 0.0059 according to the general χ^2^-test comparing A vs. non-A; see Figure 2). Table 3 summarizes the recorded ECG results.

To check whether the association between ECG-AHA classification and CD status was not influenced by confounding factors, we analyzed the data with a multi-factorial logistic regression model in addition using sex and age (in groups 12–20, 20–30, 30–40, … and 80–90) for adjustment. All three factors remain statistically significant on the 5% level with *p*-values of 0.0107 (CD-status), 0.0119 (sex), and <0.0001 (age). Within this adjusted model, the odds ratio comparing ECG-AHA B1-D with A is 3.994 assuming a nearly 4-fold higher odds in Chagas-positive compared to Chagas-negative participants.

This general observation of more severe ECG-findings in CD-positive participants aged 12 years and older was reproducible for individual ECG diagnoses as well. The latter were mostly observed in Chagas-positive patients only (see Table 3 for details).

## 4. Discussion

The indigenous communities examined in this study live in regions of high endemicity for CD. Simple living conditions like clay huts, palm roofs, unsealed floors, a close contact to livestock, low levels of education next to other factors favor CD infections and their transmission. Prevalence in the different villages ranges from 43.6% in Seminke and up to 67.4% in Sabanna de Higuerón. The applied highly specific real-time-PCR for CD showed positive results in serologically positive patients in various villages ranging from 11.4% to 39.3%. These results impressively confirm that CD is a considerable regional health issue threat for the indigenous people living in this area. The results presented here are well in line with high number of infections, as reported in a previous study [18]. Many other Latin American countries like, for example, Bolivia, suffer from high CD burdens, too [2]. However, it seems that the indigenous population in the Sierra Nevada, in particular, is even more strongly affected even compared to other South American high-endemicity settings [2,18].

The situation is aggravated by the fact that healthcare access for the Wiwa population is sparse. More than six hours walking distance are needed to reach the next health point from various Wiwa villages, but even at the health point or at the next hospital, the awareness for CD is low as diagnostic tools and education are missing. The problem is aggravated by the complexity of diagnostic protocols for CD which are applied in Colombia. The Colombian government and guidelines follow the WHO recommendations, which demands positive results in at least two different ELISAs and—in case of a discordant result—even a third positive result to classify a patient as CD positive. Only after this complex procedure is a patient is entitled for treatment [22,23]. It has recently been shown, that RT test results are comparably reliable like two ELISA results and that the results of different ELISA assays are usually matching. Accordingly, it has been suggested to ease the criteria required to entitle affected patients suffering from CD for therapy [18]; however, those suggestions are not yet established. In addition, the SARS-CoV-2 pandemic facilitated the availability of diagnostic PCR in Colombia. Essentially, all public health institutes in Colombia have a PCR machine available now and, in addition, PCR kits targeting CD, partially based on relatively new, patent protected protocols (e.g., patent number DE 2015 111 267.1; 19 January 2017: “Kann, S.; Hansen, J. Oligonukleotide und deren Verwendung.”), are available on the market to facilitate CD screening and therapy control.

To allow treatment for more potentially infected individuals and thus to avoid premature death and disability, the purpose of this study was to assess whether and in how far physical examinations, targeted questionings, and/or ECG findings are useful to guide the examiner regarding the decision on the indication for diagnostic CD testing. Early diagnosis is thereby particularly important because acute Chagas infections have the most promising therapeutic outcomes. However, patients suffering from chronic stages of the disease can have therapeutic benefit as well because treatment can prevent the further progress of the disease [12] or even lead to some clinical improvement.

The assessment of the results from the CD-specific questionnaires indicated that CD-positive patients show a significantly different profile in comparison to CD-negative individuals. So, CD-positive patients claimed symptoms in 86.1% of cases, whereas this was true for only 41.4% of the CD-negative cases. Symptoms like dizziness, chest pain, and fatigue were the most common ones in CD patients, and targeted questions for these and related symptoms could lead to a further evaluation of CD. In the physical examination, heart murmurs, rhythms problems, edema, etc. should also be seen as a hint to consider CD as an underlying disease.

The ECG assessments confirmed that CD-positive patients suffer from a variety of cardiac diatheses and that these can be observed at different stages of the disease. An odds ratio of 2.6 was calculated for pathological cardiological findings in CD-positive patients 12 years of age and older. In total, 29.6% of the infected showed any irregular finding, ranging from B to D according to the Chagas associated American Heart Association (AHA) criteria [15]. In summary, the findings result in a nearly 4-fold increase in the odds ratio comparing CD-positives with CD-negatives in an age- and sex-adjusted multi-factorial logistic regression model.

Cardiac rhythm disorders, in particular incomplete right bundle block, were the most commonly observed findings. Although such results were seen in healthy, non-infected individuals as well, they were significantly more frequent in *T. cruzi*-infected individuals in line with previous reports [15]. Consequently, an ECG should be obligatory for CD-positive individuals but in turn, it should also be used for screening purposes in the high endemicity setting even in the field to obtain a first impression on the stage and severity of the patient’s cardiac status and to indicate CD diagnosis. Because it is easy to perform an ECG in the field and the machine is transportable with a long lasting battery and not cost-intense, we recommend that an ECG screening should be included in the diagnostic algorithm for the indigenous communities in their high endemicity setting for CD in order to further improve the CD screening in this population in need.

Due to their insufficient specificity, CD-related symptoms and ECG changes cannot be the only criteria for a CD case definition. However, if they are observed in a highly endemic region for CD, they should lead to a consideration of CD as an important differential diagnosis. As a consequence, at least a rapid test (RT) for the further evaluation of the patient should be in place. In case of a positive RT result, further confirmatory testing should be initiated to make sure that the patient can receive adequate treatment, minimizing the patient’s risk of severe and life-shortening complications.

Next to its primary aim, the study also once more confirmed the previous observation that gastrointestinal infections are very common in Wiwa communities [19,20,21,24,25,26]. As the complaints disappeared after an appropriate treatment of the detected gastrointestinal pathogens (data not shown), an association with CD is unlikely. As stated above, it is likely that the high numbers of such infections are a result of poor socioeconomic conditions and a lack of clean water and sanitation.

The study has a number of limitations. For example, it would have been desirable to confirm pathologic findings from ECGs with cardiac echocardiography, cardiac parameters in peripheral blood. As there is just a single cardiologist available in the nearby zone and the distance to this specialist is still very far, it was not feasible to include such examinations in the study approach for logistical reasons and also not for capacity reasons of the cardiologist. In addition, specialized laboratories, which are able to measure cardiac parameters like pro-BNP or troponin, are far away. Their inclusion would also have required logistic and financial support that was not available for the study.

## 5. Conclusions

The indigenous Wiwa communities are severely and quantitatively relevantly affected by CD. Screenings for the disease could be improved by including RT- and ECG-based algorithms. CD-related complaints should be routinely asked for as an element of the medical history in the local high endemicity settings. Suggestive findings should trigger a transfer of suspected cases to the next CD diagnostic center. Countermeasures, awareness, and controls need to be started and/or expanded.

## Figures and Tables

**Figure 1 tropicalmed-08-00297-f001:**
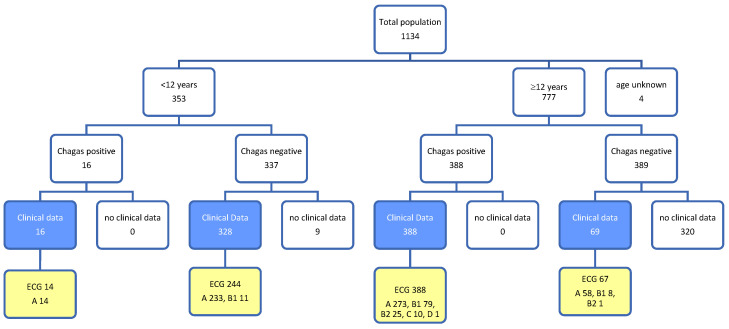
Overview on the volunteers regarding age, CD status (Chagas positive means at least positive in two ELISAs), clinical data (questionnaire including physical examination; complete data sets shaded blue) and ECG data (complete data sets shaded yellow).

**Figure 2 tropicalmed-08-00297-f002:**
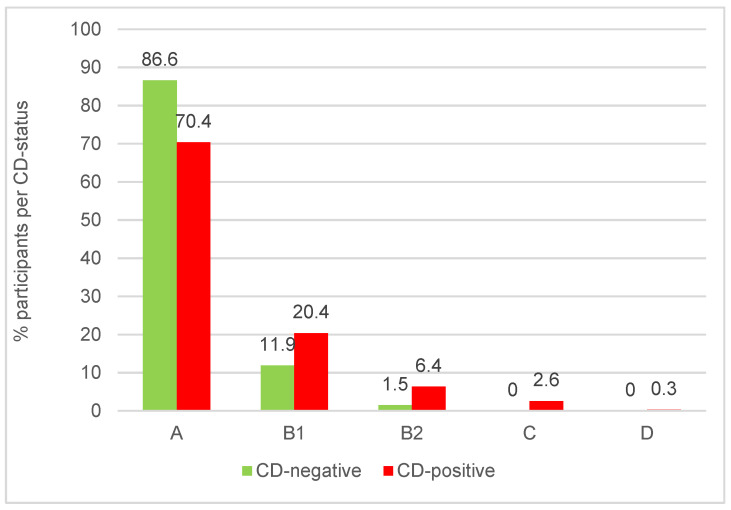
ECG-AHA classification by CD status in %. Data from *n* = 455 participants aged 12 years and older are shown.

**Table 1 tropicalmed-08-00297-t001:** CD-positive cases (positive in serology or PCR or both) as well as CD-negative participants distributed by village.

Village	CD-Positive		CD-Negative		All
	*n*	%	*n*	%	*n*
Sabana de Higuerón	58	67.4	28	32.6	86
Potrerito	28	62.2	17	37.8	45
Dungakare	22	57.9	16	42.1	38
Ahuyamal	47	54.7	39	45.3	86
Ashintukwa	50	47.2	56	52.8	106
Cherua	44	46.8	50	53.2	94
Surimena	29	46.0	34	54.0	63
Tezhumake	78	45.1	95	54.9	173
Seminke	48	43.6	62	56.4	110
All	404	50.4	397	49.6	801

**Table 2 tropicalmed-08-00297-t002:** CD-positive (positive in serology or PCR or both) and CD-negative participants’ complaints by CD-status and age groups.

Symptoms	CD-Positive		CD-Negative	
	*n*	%	*n*	%
participants younger than 12 years				
yes	7	43.8	55	16.8
no	9	56.3	273	83.2
Participants being 12 years of age and older				
yes	164	42.3	17	24.6
no	224	57.7	52	75.4

**Table 3 tropicalmed-08-00297-t003:** Total number of ECG-diagnoses by CD status in participants aged 12 years and older.

ECG-Diagnosis	CD-Negative		CD-Positive		All
	*n*	%	*n*	%	*n*
No pathological findings	57	83.8	254	60.3	311
Rhythm-associated findings					
Sinus bradycarida	8	11.8	84	20.0	92
Partial RSB	0	0	26	6.2	26
Total RSB	0	0	20	4.8	20
AV Bradycardia	0	0	3	0.7	3
Partial LSB	0	0	3	0.7	3
Deformed atrial extrasystolic activity	0	0	2	0.5	2
Sinus tachycardia	0	0	1	0.2	1
AV Block	0	0	1	0.2	1
Extrasystolic ventricular impulse	0	0	1	0.2	1
Short PR Interval	0	0	1	0.2	1
WPW Syndrome	0	0	1	0.2	1
Left anterior hemiblock	1	1.5	1	0.2	2
Intraventricula conduction block	0	0	1	0.2	1
Sinus arrythmia	1	1.5	0	0	1
Heart muscle-associated findings					
Left deviation of axis	1	1.5	6	1.4	7
Right deviation of axis	0	0	6	1.4	6
Old myocardial lesion	0	0	5	1.2	5
Left atrial hypertrophy	0	0	1	0.2	1
Left ventricular hypertrophy	0	0	1	0.2	1
Biventricular hypertrophy	0	0	1	0.2	1
Others					
Pacemaker wave	0	0	1	0.2	1
Low voltage	0	0	1	0.2	1
All	68	100.0	421	100.0	489

RSB = right bundle block, LSB = left bundle block, AV = atrioventricular, WPW = Wolff Parkinson White Syndrome. PR interval = distance between the P and the R wave.

## Data Availability

The data presented in this study are available on request from the corresponding author.

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
