# Peer review of "Chagas Disease: Medical and ECG Related Findings in an Indigenous Population in Colombia"

_tropicalmed, 2023, doi:10.3390/tropicalmed8060297_

Round 1

Reviewer 1 Report

This study shows Chagas disease prevalence among ages in Wiwa villages in Magdalena, Colombia. Authors performed several tests: rapid-test, real-time PCR, Elisa and ECG along with symptom records for presenting their results. I think this is a great study given the focus population and the number of individuals involved. Not a lot of studies consider such rural populations, which brings us closer to the true conditions the continent faced when Chagas disease transmission began. 

I have only one concern which is that the ecological aspects of Chagas disease transmission in the region is barely discussed. I suggest it is important to mention the main vector species in the area and its ecology, such as blood meal preferences, activity peak times, habitat, etc. Second, what are the most important hosts in the area (sylvatic or domestic mammals?). Third, how villages vary in their habits, is this a differential risk factor for CD in the Wiwa community? I believe that by including a() paragraph(s) considering the socio-ecological factors that led such high transmission in the community, the paper would be more sound.

Additional comments:

line 58: typo encepahlitis

line 67: typo indivdiduals

line 121: typo sympotoms

Author Response

Reviewer 1

This study shows Chagas disease prevalence among ages in Wiwa villages in Magdalena, Colombia. Authors performed several tests: rapid-test, real-time PCR, Elisa and ECG along with symptom records for presenting their results. I think this is a great study given the focus population and the number of individuals involved. Not a lot of studies consider such rural populations, which brings us closer to the true conditions the continent faced when Chagas disease transmission began. 

I have only one concern which is that the ecological aspects of Chagas disease transmission in the region is barely discussed. I suggest it is important to mention the main vector species in the area and its ecology, such as blood meal preferences, activity peak times, habitat, etc. Second, what are the most important hosts in the area (sylvatic or domestic mammals?). Third, how villages vary in their habits, is this a differential risk factor for CD in the Wiwa community? I believe that by including a() paragraph(s) considering the socio-ecological factors that led such high transmission in the community, the paper would be more sound.

Additional comments:

line 58: typo encepahlitis

line 67: typo indivdiduals

line 121: typo sympotoms

Answer: Thank you very much for the valuable hints. We have corrected the typing mistakes and have added a socio ecological section in the introduction part according to your suggestions.

It is now: “The region examined has many of the potentially transmitting vectors: One of the main vectors in the area examined is Rhodnius prolixus. It is a member of the family Reduviidae and transmits CD to humans and other mammals. R. prolixus is usually present in rural areas and feeds on the blood of animals such as birds, rodents, and opossums. It is attracted by the carbon dioxide that mammals exhale and is typically active at night, when the hosts are asleep [7,8]. As the Wiwas live close together with their life stock and as the housing and climate conditions are favorable for the Triatomines (palm roofs, mud walls, etc.), transmission to humans is common [9]. Another locally highly prevalent triatomine is Triatoma dimidiata, which transmits CD as well. It is also attracted to carbon dioxide, but in addition to heat and moisture, showing its perfect adaption to the climate conditions found in the Sierra Nevada de Santa Marta. Also, other triatomes can be observed in the region, e.g., T. maculata, T. infestans, R. pallenscens (the so called “palm tree triatomine”), Panstrongylus geniculatus, however, those play only a subordinate role so far [8,10,11]. Other reasons for the observed high number of infections in the indigenous populations, next to the poor socio-economic living conditions, comprise a lack of countermeasures like fumigations, an insufficient knowledge and awareness of the disease, and a sparse access to surveillance and prevention programs, e.g., due to their retracted living places. Health care is also limited and just consists out of a nurse in a health point with irregular consultation days without options of diagnosing T. cruzi-infections.“

Reviewer 2 Report

Dear authors,

The manuscript presents relevant information for the diagnosis of Chagas disease. I will leave below some small considerations to contribute to the work:

Line 32: I believe that a sentence mentioning the etiological agent of Chagas disease, pointing  the taxonomic classification, would be relevant in the first sentence of the introduction, for example:

“Chagas disease (CD), caused by the protozoan Trypanosoma cruzi (Chagas, 1909) (Kinetoplastida, Trypanosomatidae), is a high burden in the indigenous tribe called Wiwa.”

Line 43: Triatomines can be written in lower case and must be accompanied by the taxonomic classification at the first mention of these insects:

“... triatomines (Hemiptera, Reduviidae, Triatominae)”.

Furthermore, still on line 43, the parasite is transmitted, not the disease.

Line 45: I believe after “The local main mode of transmission is via Triatomines that suck human blood, thereby leaving infected stool close to the stich point.” a sentence mentioning the most common species in the region would be interesting, highlighting those of greater vectorial importance.

Line 396: The section 6 title is at the end of the conclusion.

References need to be revised, as they are not in accordance with the journal's guidelines (https://www.mdpi.com/journal/tropicalmed/instructions#references)

Best regards.

Author Response

Dear authors,

The manuscript presents relevant information for the diagnosis of Chagas disease. I will leave below some small considerations to contribute to the work:

Line 32: I believe that a sentence mentioning the etiological agent of Chagas disease, pointing  the taxonomic classification, would be relevant in the first sentence of the introduction, for example:

“Chagas disease (CD), caused by the protozoan Trypanosoma cruzi (Chagas, 1909) (Kinetoplastida, Trypanosomatidae), is a high burden in the indigenous tribe called Wiwa.”

Answer: Thank you for that advice, we have corrected it.

Line 43: Triatomines can be written in lower case and must be accompanied by the taxonomic classification at the first mention of these insects:

“... triatomines (Hemiptera, Reduviidae, Triatominae)”.

Answer: We have added this.

Furthermore, still on line 43, the parasite is transmitted, not the disease.

Answer: The sentence/section was reorganized, so that also this was corrected.

Line 45: I believe after “The local main mode of transmission is via Triatomines that suck human blood, thereby leaving infected stool close to the stich point.” a sentence mentioning the most common species in the region would be interesting, highlighting those of greater vectorial importance.

Answer: Due to the comment of Reviewer I, the sections was expanded and main vectors were named. It is now: “The region examined has many of the potentially transmitting vectors: One of the main vectors in the area examined is Rhodnius prolixus. It is a member of the family Reduviidae and transmits CD to humans and other mammals. R. prolixus is usually present in rural areas and feeds on the blood of animals such as birds, rodents, and opossums. It is attracted by the carbon dioxide that mammals exhale and is typically active at night, when the hosts are asleep [7,8]. As the Wiwas live close together with their life stock and as the housing and climate conditions are favorable for the Triatomines (palm roofs, mud walls, etc.), transmission to humans is common [9]. Another locally highly prevalent triatomine is Triatoma dimidiata, which transmits CD as well. It is also attracted to carbon dioxide, but in addition to heat and moisture, showing its perfect adaption to the climate conditions found in the Sierra Nevada de Santa Marta. Also, other triatomes can be observed in the region, e.g., T. maculata, T. infestans, R. pallenscens (the so called “palm tree triatomine”), Panstrongylus geniculatus, however, those play only a subordinate role so far [8,10,11]. Other reasons for the observed high number of infections in the indigenous populations, next to the poor socio-economic living conditions, comprise a lack of countermeasures like fumigations, an insufficient knowledge and awareness of the disease, and a sparse access to surveillance and prevention programs, e.g., due to their retracted living places. Health care is also limited and just consists out of a nurse in a health point with irregular consultation days without options of diagnosing T. cruzi-infections.“

Line 396: The section 6 title is at the end of the conclusion.

Answer: Thank you very much, yes, we have deleted this.

References need to be revised, as they are not in accordance with the journal's guidelines (https://www.mdpi.com/journal/tropicalmed/instructions#references)

Answer: We have worked over the references and hope, they are all as requested now.

Reviewer 3 Report

The objective of this work is to associate clinical conditions with possible alterations in the ECG. The work is relevant but still preliminary and needs to improve the writing of unclear results and increase the number of patients studied.

Some comments and suggestions to improve the work:

The methodology states that there are two groups, one with children under 12 years old and the other with people over 12 years old. But it doesn't say the real range in over 12 years. This needs to be made clear in the text.

The results are divided into under 12 years old and over 12 years old. This gap in the second group can be very extensive. I suggest subdividing the groups every ten years or by children, young people, adults and seniors, being informed of the age range for each group. The results as presented do not give the real notion of which group we are talking about. So much so that body measurements and blood pressure according to the author were not added due to the large interval and age difference in the group over 12 years. This makes the results inaccurate.

It is necessary to correlate the age of the patients with the positivity in the figures and graphs. Especially when we look at the results in Figure 2 and Table 3, which list the main clinical pathologies found and the ECG results. How to know if some symptoms are not related to old age? It is important to state whether the results were significant or not and the p value in each analysis. I suggest reviewing and rewriting these results more clearly.

A discussão precisa levar em consideração os dados encontrados com a idade do paciente e discutir esses achados de maneira mais profunda com a literatura. 

Author Response

The objective of this work is to associate clinical conditions with possible alterations in the ECG. The work is relevant but still preliminary and needs to improve the writing of unclear results and increase the number of patients studied.

Some comments and suggestions to improve the work:

The methodology states that there are two groups, one with children under 12 years old and the other with people over 12 years old. But it doesn't say the real range in over 12 years. This needs to be made clear in the text.

Answer: Thanks for this comment. The two groups stated are parts of the study design by definition (see details in section 2.3). Within this, participants less than 12 years were defined as children, while participants 12 years and older were defined as adults. To make this much clearer we re-drafted the description of the study design and give additional information on the structure of the "adult study collective" by adding some additional percentiles within the result section.

The results are divided into under 12 years old and over 12 years old. This gap in the second group can be very extensive. I suggest subdividing the groups every ten years or by children, young people, adults and seniors, being informed of the age range for each group. The results as presented do not give the real notion of which group we are talking about. So much so that body measurements and blood pressure according to the author were not added due to the large interval and age difference in the group over 12 years. This makes the results inaccurate.

Answer: We really are in line with the reviewer's comment that the second age group has a huge spread in age. As requested we therefore have added some additional information in the result section describing the age distribution of the adult group in more detail (see answer on comment above).

Answer: However, we respectfully disagree that our results are inaccurate by do not taking age into account in more detail. First, following the design of the study, separated analyses by age group were done. Second, for children younger than 12 years, only few ECG symptoms were reported. Therefore, most of the analyses are concentrated on the older age group only. Further, discussing that body measurement or blood pressure is correlated with age was no new data, as these relations are known for many years and were published over and over so that it is already seen as evidence based data, so that we don’t see further necessity of additional addressing this. Accordingly, we decided to add multifactorial (logistic) analyses dealing with figure 2, but preserve the data description in table 3 due to the small number of findings here. In addition, we have extended the discussion on this point.

It is necessary to correlate the age of the patients with the positivity in the figures and graphs. Especially when we look at the results in Figure 2 and Table 3, which list the main clinical pathologies found and the ECG results. How to know if some symptoms are not related to old age? It is important to state whether the results were significant or not and the p value in each analysis. I suggest reviewing and rewriting these results more clearly.

Answer: Please see our answer of the above query.

A discussão precisa levar em consideração os dados encontrados com a idade do paciente e discutir esses achados de maneira mais profunda com a literatura. 

Answer: As stated above, the age-factor was addressed and included in the discussion.

Round 2

Reviewer 3 Report

All questions were answered by the authors. Being a preliminary work, I think that the changes made to the text meet the requirements.